# Synthesis and Thermo-Responsive Behavior of Poly(*N*-isopropylacrylamide)-*b*-Poly(*N*-vinylisobutyramide) Diblock Copolymer

**DOI:** 10.3390/polym16060830

**Published:** 2024-03-18

**Authors:** Jun Hyok Yoon, Taehyoung Kim, Myungeun Seo, Sang Youl Kim

**Affiliations:** 1Department of Chemistry, Korea Advanced Institute of Science & Technology (KAIST), Daejeon 34141, Republic of Korea; yoonjun5@kaist.ac.kr (J.H.Y.); thkim93@kaist.ac.kr (T.K.); seomyungeun@kaist.ac.kr (M.S.); 2KAIST Institute for Nanocentry, Korea Advanced Institute of Science & Technology (KAIST), Daejeon 34141, Republic of Korea

**Keywords:** thermo-responsive polymer, lower critical solution temperature (LCST), block copolymer, controlled radical polymerization, reversible addition–fragmentation chain transfer polymerization (RAFT), self-assembly, polymer synthesis

## Abstract

Thermo-responsive diblock copolymer, poly(*N*-isopropylacrylamide)-*block*-poly(*N*-vinylisobutyramide) was synthesized via switchable reversible addition–fragmentation chain transfer (RAFT) polymerization and its thermal transition behavior was studied. Poly(*N*-vinylisobutyramide) (PNVIBA), a structural isomer of poly(*N*-isopropylacrylamide) (PNIPAM) shows a thermo-response character but with a higher lower critical solution temperature (LCST) than PNIPAM. The chain extension of the PNVIBA block from the PNIPAM block proceeded in a controlled manner with a switchable chain transfer reagent, methyl 2-[methyl(4-pyridinyl)carbamothioylthio]propionate. In an aqueous solution, the diblock copolymer shows a thermo-responsive behavior but with a single LCST close to the LCST of PNVIBA, indicating that the interaction between the PNIPAM segment and the PNVIBA segment leads to cooperative aggregation during the self-assembly induced phase separation of the diblock copolymer in solution. Above the LCST of the PNIPAM block, the polymer chains begin to collapse, forming small aggregates, but further aggregation stumbled due to the PNVIBA segment of the diblock copolymer. However, as the temperature approached the LCST of the PNVIBA block, larger aggregates composed of clusters of small aggregates formed, resulting in an opaque solution.

## 1. Introduction

Thermo-responsive polymers are a class of smart polymers that undergo reversible changes in their physical properties when exposed to temperature change. Thermo-responsive polymer solutions are characterized by a volume phase transition triggered by temperature, resulting in a change in the solvation state of the polymer [1,2]. When the solution of a polymer becomes an immiscible mixture for all compositions over a certain temperature, that temperature is referred to as the lower critical solution temperature (LCST) [3,4,5]. The polymers showing LCST behavior in an aqueous solution have a moiety capable of forming hydrogen bonds with water such as oligo(ethylene glycol)s and second or tertiary amines including Poly(*N*-alkylacrylamide)s and poly(*N*-vinylalkylamide)s [6,7,8]. The driving force of the transition arises from the equilibrium between hydrophilic and hydrophobic interaction segments in the polymer chain. One of the most well-known examples of a thermo-responsive polymer is poly(*N*-isopropylacrylamide) (PNIPAM). PNIPAM exhibits LCST behavior that is soluble in water below 32 °C but becomes insoluble and precipitates out of the solution when the temperature is raised above its LCST. PNIPAM is widely used because its thermo-responsive transition is less dependent on chemical environments such as pH, and salt concentrations [9,10,11,12,13,14,15]. Due to the unique characteristics exhibited by PNIPAM, PNIPAM based copolymer have a broad spectrum of research in self-assembly and stimuliresponsive systems [16,17,18,19]. High concentrations above 10% of PNIPAM and PNIPAM-based copolymer aqueous solutions undergo reversible gelation [20,21]. Because the LCST of PNIPAM is lower than the human body temperature (~37 °C), the PNIPAM-based polymer system can exist as a gel state around human body temperature [22,23].

The thermo-responsive properties of polymers can be tailored by modifying the polymer structure and composition. Diblock copolymers can be used to modulate LCST behavior, and one of the advantages of using diblock copolymers with LCST behavior is the ability to fine-tune the LCST by selecting appropriate polymer blocks and their relative proportions [24,25]. The diblock copolymers consisting of PNIPAM show interesting thermo-responsive properties with high potential for various applications, including drug delivery, tissue engineering, nanoreactors, and wearable devices [26,27,28,29]. Among the structural isomers of poly(*N*-isopropylacrylamide) including poly(*N*-ethyl methacrylamide), poly(*N*,*N*-ethyl methyl acrylamide), poly(*N*-n-propyl acrylamide), poly(*N*-isopropyl acrylamide), poly(*N*-vinyl isobutyramide), poly(2-n-propyl-2-oxazoline), poly(2-isopropyl-2-oxazoline), polyleucine, and polyisoleucine, we select poly(*N*-vinylisobutyramide) because the structural difference is minimal. They consist of amide groups. The only difference is the direction of amide linkage in the side chain. However, this small difference gives rise to the observable differences in LCST (32 °C and 39 °C, respectively), indicating the different solvation behavior with water [30,31]. We are interested in whether this small structural difference is large enough for sequential aggregation behavior in water.

Reversible addition–fragmentation chain transfer (RAFT) polymerization is compatible with a wide range of monomers, making it suitable for synthesizing various block polymers [32,33]. However, NVIBA monomer which can be prepared from *N*-vinylformamide (NVF) at mild conditions [34] is categorized as “less-activated” monomers (LAMs) for RAFT polymerization because it has lone pair electrons adjacent to its vinyl group compared to “more-activated” monomers (MAMs) which have double bond conjugated to an aromatic ring (e.g., styrene) or a carbonyl group (e.g., methylmethacrylate). As the propagating radical of LAMs has low stability and side reactions, these monomers behave as poor homolytic leaving groups in radical polymerization. Therefore, typical RAFT reagents, such as thiocarbonylthio compounds, that can polymerize MAMs in a controlled manner are not suitable for the RAFT polymerization of LAMs. The low stability of LAM radicals can be compensated with xanthates or dithiocarbamates RAFT agents, but they are ineffective in the controlled polymerization of MAMs [33,35,36].

Because of the radical reactivity inherent in NVIBA, the synthesis of PNVIBA and its copolymers has predominantly relied on free radical polymerization, and most reports use PNIVBA as a building segment in chemically crosslinked hydrogels [37,38,39,40]. The synthesis of diblock copolymers compromising LAM and MAM blocks requires a RAFT agent of which reactivity can be modulated for both LAMs and MAMs [41,42,43,44,45]. The switchable RAFT agent, *N*-(4-pyridinyl)-*N*-methyldithiocarbamate is effective for RAFT polymerization of LAMs. Interestingly, it becomes effective for MAMs when it is protonated with a strong acid, allowing a direct synthesis of polyMAM-*block*-polyLAM with controlled molecular weight and molecular weight distributions [35,43]. However, to the best of our knowledge, synthesis as well as self-assembly behavior of poly(*N*-isopropylacrylamide)-block-poly(*N*-vinylbutyramide) (PNIPAM-*b*-PNVIBA) has not been reported. Herein, we report the synthesis and thermo-responsive behavior of the PNIPAM-*b*-PNVIBA diblock copolymer.

## 2. Materials and Methods

### 2.1. Materials

Methyl 2-[methyl(4-pyridinyl)carbamothioylthio]propionate (MMPCP, Sigma-Aldrich, Saint Louis, MO, USA, 97%), 2-cyanopropan-2-yl-*N*-methyl-*N*-(pyridin-4-yl)carbamodithioate (CPMPC, Sigma-Aldrich, 97%), 4-dimethylaminopyridine (DMAP, Sigma-Aldrich, 99%), and trimethylamine (anhydrous, Sigma-Aldrich, 99%) were purchased. *N*-isopropylacrylamide (NIPAM, TCI, Tokyo, Japan, 98%) was recrystallized from n-hexane two times. *N*-Vinylformamide (NVF, TCI, 98%), and isobutyryl chloride (TCI, 98%) were distillated with reduced pressure to remove water and other impurities. The *N*-vinylisobutyramide (NVIBA) was synthesized according to the following procedure [34]. 2,2′-Azobis(2-metylpropionitrile) (AIBN, Junsei, Tokyo, Japan, 98%) was recrystallized from methanol before use. 1,4-Dioxane (anhydrous, Sigma Aldrich, 99%), dimethyl sulfoxide (DMSO, anhydrous, Sigma-Aldrich, 99%), and tetrahydrofuran (THF, anhydrous, Sigma-Aldrich, 99%) were used as received.

### 2.2. Characterizing Methods

^1^H nuclear magnetic resonance (NMR) spectroscopy was taken on Bruker Fourier Transfer AVHD400 spectrometer (Bruker, Billerica, MA, USA). Chemical shifts are expressed in parts per million (ppm) using DMSO-*d*_6_, and D_2_O solvent protons as references. High-temperature ^1^H nuclear magnetic resonance experiment was conducted at 100 °C with Bruker Fourier Transfer AVHD400 spectrometer. Gel permeation chromatography (GPC) was performed in *N*,*N*-dimethylformamide (DMF) eluent at 45 °C with a flow rate of 1 mL/min on Agilent 1260 Infinity system (Santa Clara, CA, USA). The instrument was equipped with a 1260 refractive index detector. The molecular weight of the polymers was calculated relative to linear poly(styrene) standards (Agilent Polystyrene Medium EasiVials, Agilent, Santa Clara, CA, USA, M_w_ by Light scattering: 208,300, 28,100, 3610, 452). DLS measurement was conducted using NanoBrook Omni (Brookhaven Instruments Corp., Holtsville, NY, USA) at a wavelength of 658 nm. The scattering angle used for the measurements was 90 degrees. For the DLS measurements, the temperature range was 25 to 55 °C, and the measurements were conducted at 1 °C intervals. Each measurement was equilibrated for 60 s and measured for 120 s, and the measurement cycle was repeated 3 times. The CONTIN approximation was used to convert the diffusion coefficient into the hydrodynamic diameter (*D*_h_). For the DLS measurements, the polymer solution concentration was 0.5 mg/mL. Zeta potential measurement was performed on Malvern Zetasizer nano ZS at a wavelength of 632 nm with Malvern DTS1070 disposable folded capillary cell. For the zeta potential measurement, the temperature range was 25 to 55 °C, and each measurement was conducted at 1 °C intervals. Each measurement was equilibrated for 60 s. The polymer solution concentration was 0.5 mg/mL. UV-vis spectroscopy was carried out on Shimadzu UV-2600 equipped with a temperature controller system. The scan wavelength was set at 550 nm. The polymer solution sample was equilibrated at 25 °C for 300 s before the measurement and ramped at a rate of 0.1 °C/min. For UV-vis studies, cloud point (CP) is defined as a temperature when the transmittance becomes 0.5 [46,47]. And the ‘temperature of transmittance change’ is defined as the temperature at 99% transmittance. For all UV-vis studies, the polymer solution concentration was 0.5 mg/mL. A scanning electron microscopy (SEM) study was performed on a FEI Magellan 400 (FEI, Hillsboro, OR, USA). SEM sample was prepared by spin coating of 0.5 mg/mL aqueous polymer solution on Si wafer. Thermogravimetric analysis (TGA) was performed on a Netzsch TG 209 F1 Libra high resolution TGA (Netzsch, Selb, Germany). The TGA measurements were conducted at a heating rate of 10 °C/min under nitrogen conditions, and the measurement temperature range was 100 to 600 °C. Differential scanning calorimetry (DSC) was performed by TA instruments DSC25 differential scanning calorimeter (TA instruments, New Castle, DE, USA). The glass transition temperature (*T*_g_) values of the polymers were obtained with a DSC instrument at a heating rate of 10 °C/min under N_2_ conditions, and the measurement temperature range was −10 to 250 °C. The DSC measurement data were obtained during the second heating cycle.

### 2.3. Polymer Synthesis

Synthesis of PNIPAM_128_ macro-RAFT agent. In a dry Schlenk flask, NIPAM (1.6974 g, 15 mmol), MMPCP (0.02704 g, 0.1 mmol), AIBN (3.3 mg, 0.02 mmol), and TsOH(monohydrate) (20.9 mg, 0.11 mmol) were placed and dissolved in 1,4-dioxane (4 mL). The flask was sealed, and then the solution was degassed by repeated freeze, pump, and thaw cycles at least three times. Then, the reaction flask was filled with nitrogen gas. Polymerization was started by placing the flask in an oil bath at 60 °C for 4 h. The polymerization was stopped by diluting and cooling the solution in an ice-water bath. The polymer was isolated by precipitation into diethyl ether and isolated by filtration. The product was dried in vacuo overnight. *M*_n_: 1.44 × 10^4^ g/mol, PDI: 1.19, ^1^H NMR (400 MHz, CDCl_3_) δ 8.73, 7.41, 7.26–6.52, 3.87, 2.04, 1.78–1.20, 1.10.

Synthesis of PNIPAM_258_ macro-RAFT agent. In a dry Schlenk flask, NIPAM (2.829 g, 25 mmol), MMPCP (0.02704 g, 0.1 mmol), AIBN (3.28 mg, 0.02 mmol), and TsOH(monohydrate) (20.9 mg, 0.11 mmol) were placed and dissolved in 1,4-dioxane (4 mL). The flask was sealed, and then the solution was degassed by repeated freeze, pump, and thaw cycles at least three times. Then, the reaction flask was filled with nitrogen gas. Polymerization was started by placing the flask in an oil bath at 60 °C for 4 h. The polymerization was stopped by diluting and cooling the solution in an ice-water bath. The polymer was isolated by precipitation into diethyl ether and isolated by filtration. The product was dried in vacuo overnight. *M*_n_: 2.95 × 10^4^ g/mol, PDI: 1.16, ^1^H NMR (400 MHz, DMSO-*d_6_*) δ 8.72, 7.52, 7.28–6.48, 3.88, 2.03, 1.68, 1.60–1.19, 1.10

Synthesis of PNIPAM_128_-*b*-PNVIBA_93_ diblock copolymer. In a dry Schlenk flask, a solution of NVIBA (0.566 g, 5 mmol), PNIPAM_128_ macro-RAFT agent (0.296 g, 0.02 mmol), AIBN (0.82 mg, 0.005 mmol), and anhydrous DMSO (1.5 mL) was prepared. The flask was sealed, and then the solution was degassed by repeated freeze, pump, and thaw cycles at least three times. Then, the reaction flask was filled with nitrogen gas. Polymerization was started by placing the flask in an oil bath at 70 °C for 16 h. The polymerization was stopped by diluting the solution and cooling in a water bath. The polymer was isolated by precipitation into diethyl ether and isolated by filtration. The product was dried in vacuo overnight. *M*_n_: 2.53 × 10^4^ g/mol. PDI: 1.29, ^1^H NMR (400 MHz, DMSO-*d_6_*_,_ 100 °C) δ 7.54, 7.48–6.49, 3.88, 3.69, 2.34, 2.03, 1.79–1.19, 1.10, 1.04.

PNIPAM_258_-*b*-PNVIBA_67_ diblock copolymer. In a dry Schlenk flask, a solution of NVIBA (0.8487 g, 7.5 mmol), PNIPAM_258_ macro-RAFT agent (0.885 g, 0.03 mmol), AIBN (1.23 mg, 0.0075 mmol), and anhydrous DMSO (2 mL) was prepared. The flask was sealed, and then the solution was degassed by repeated freeze, pump, and thaw cycles at least three times. Then, the reaction flask was filled with nitrogen gas. Polymerization was started by placing the flask in an oil bath at 70 °C for 16 h. The polymerization was stopped by diluting the solution and cooling in a water bath. The polymer was isolated by precipitation into diethyl ether and isolated by filtration. The product was dried in vacuo overnight. *M*_n_: 3.71 × 10^4^ g/mol, PDI: 1.25, ^1^H NMR (400 MHz, DMSO-*d*_6_, 100 °C) δ 7.54, 7.40–6.39, 3.87, 3.68, 2.34, 2.03, 1.78–1.19, 1.09, 1.04

## 3. Results and Discussion

### 3.1. Synthesis of PNIPAM-b-PNVIBA Diblock Copolymers

To synthesize poly(*N*-isopropylacrylamide)-*block*-poly(*N*-vinylisobutyramide) (PNIPAM-*b*-PNVIBA) diblock copolymer, *N*-vinylisobutyramide (NVIBA) monomer was prepared by reacting *N*-vinylformaldehyde with isobutyryl chloride and NaOH treatment according to the reported procedure [34]. ^1^H-NMR analysis of the reaction product supported the successful synthesis of NVIBA (Appendix A). Due to the reactivity difference between NVIBA and *N*-isopropylacrylamide (NIPAM) monomers, the PNIPAM block consisting of more activated monomer NIPAM was prepared prior to the PNVIBA block. RAFT polymerization of NIPAM with switchable RAFT agent methyl 2-[methyl(4-pyridinyl)carbamothioylthio]propionate (MMPCP) in an acidic condition produced well-controlled PNIPAM macro-RAFT agent. The protonated switchable RAFT agent MMPCP was effective in controlled RAFT polymerization of NIPAM monomers. The synthesis of PNIPAM-*b*-PNVIBA was carried out after the neutralization of the PNIPAM macro-RAFT agent with DMAP followed by chain extension with NVIBA (Figure 1).

The molecular weight of PNIPAM macro-RAFT agent was determined by end group analysis as shown in Figure 1a. The end group analysis was performed by comparing ^1^H NMR integration values of the corresponding peaks; the proton peaks of the pyridine ring end group (a, δ = 8.7 ppm) and ^1^H peak of the isopropyl proton of NIPAM repeating unit (2, δ = 3.9 ppm). The number average degree of polymerization (DP) is 128, and the molecular weight of the PNIPAM macro-RAFT agent is 14,800 g/mol. Also, PNIPAM-*b*-PNIVBA diblock copolymer was analyzed by ^1^H NMR spectroscopy. To obtain the number average DP of PNVIBA block, block ratio analysis between PNIPAM and NVIBA blocks was conducted by comparing the integration values of the corresponding peaks. However, due to the structural similarity of PNVIBA to PNIPAM, most of the PNVIBA peaks overlapped with the PNIPAM peaks. To avoid this problem, we performed a high-temperature ^1^H NMR study (Figure 1b). When the temperature was elevated to 100 °C, the isopropyl proton of NVIBA units and vinyl backbone proton adjacent to acrylic carbon of NIPAM units were resolved, enabling the analysis of the block ratio. The number average DP of PNVIBA block is 93, and the number average molecular weight of the PNIPAM-*b*-PNVIBA diblock copolymer is 25,300 g/mol. The molecular weight distribution was characterized by gel permeation chromatography (GPC) with DMF eluent. The molecular weight distribution of the PNIPAM-*b*-PNVIBA diblock copolymer becomes broader than that of the PNIPAM macro-RAFT agent (D = 1.19), but it still maintains an unimodal distribution with a polydispersity index of less than 1.3 (Figure 2, and Table 1). For other polymers, the same analysis process was conducted (See Appendix A for the details). Thermal analyses of the PNIPAM macro-RAFT agent and PNIPAM-*b*-PNVIBA diblock copolymers were carried out to observe the thermal stability of the PNIPAM macro-RAFT agent and the miscibility of the two blocks, PNIPAM and PNVIBA blocks, of the diblock copolymers. The decomposition of PNIPAM macro-RAFT agent began at over 250 °C and reached the 5% weight loss around 300 °C, while the diblock copolymer showed that the 5 weight % loss occurred around 325 °C (Appendix A). DSC curves of PNIPAM-*b*-PNVIBA diblock copolymers shown in Appendix A revealed the single glass transition temperature, indicating that PNIPAM and PNVIBA blocks are miscible.

### 3.2. Thermo-Responsive Properties

Thermo-responsive behavior of poly(*N*-isopropylacrylamide)-*block*-poly(*N*-vinylbutyramide) (PNIPAM-*b*-PNVIBA) was investigated by observing the transmittance change of the aqueous solution of the PNIPAM-*b*-PNVIBA diblock copolymer with UV/vis spectrophotometer. The lower critical solution temperature (LCST) of the PNIPAM homopolymer (PNIPAM macro-RAFT agent) and diblock copolymer, PNIPAM_128_-*b*-PNVIBA_93_ were measured. Figure 3 shows the temperature-dependent transmittance change of 0.5 mg/mL polymer aqueous solution with temperatures ranging from 25 °C to 55 °C. PNIPAM_128_ macro-RAFT agent shows that transmittance change begins at 32.6 °C, and reaches the CP at 35.2 °C. This LCST is a little higher than the previously reported LCST of PNIPAM (32 °C) [48,49] presumably because of the end groups (methyl 2-[methyl(4-pyridinyl)carbamothioylthio]propyl group) [50].

PNIPAM_128_-*b*-PNVIBA_93_ diblock copolymer aqueous solution shows that transmittance change starts at 35.4 °C and reaches the cloud point at 36.9 °C, revealing a higher thermal transition temperature than the PNIPAM homopolymer (PNIPAM_128_ macro-RAFT agent) but a lower LCST than PNVIBA of 39 °C in water. The relatively more hydrophilic PNVIBA block interacts more strongly with water than the PNIPAM at the temperature over LCST of PNIPAM. Compared to the PNIPAM homopolymer, overall interaction between water and diblock copolymer increases due to the covalently connected two blocks, PNIPAM and PNVIBA, resulting in delayed aggregation of the PNIPAM block. However, the thermal transition of the PNVIBA block was promoted by the aggregation of the PNIPAM block above the LCST. Also, in a moderate change in pH, PNIPAM-*b*-PNVIBA diblock copolymer does not show a significant change in CP. The temperature-dependent change of transmittance of PNIPAM_128_-*b*-PNVIBA_93_ diblock copolymer solution was monitored by UV-visible spectrophotometer at various pHs (See Appendix A). At acidic conditions, the cloud point slightly decreased, while the cloud point shifted to a higher temperature under basic conditions. However, the cloud point change under the weak acid/base condition employed in this study was quite small (less than ±1 °C).

DLS experiment was conducted to check the change of hydrodynamic diameter (*D*_h_) of PNIPAM homopolymer and PNIPAM-*b*-PNVIBA diblock copolymer nanoparticles according to the temperature elevation (Figure 4). At room temperature, neither the polymer solutions exhibit any distinct assembled structures. While the PNIPAM hompolymer (PNIPAM_128_ macro-RAFT agent) solution shows a rapid increase in the *D*_h_ above LCST of the PNIPAM (35.2 °C), the PNIPAM-*b*-PNVIBA diblock copolymer solution shows a little and slow increase in *D*_h_ until the temperature reaches around 37 °C. However, above the CP of diblock copolymer (36.9 °C), the PNIPAM-*b*-PNVIBA diblock copolymer particle shows a dramatic size increase up to the LCST of PNIVIBA (39 °C). Then, the large particles shrink at a temperature higher than the LCST of PNVIBA. At this temperature, the diblock copolymer completely loses solubility in water, and the residual water molecules in the loose aggregate of PNIPAM-*b*-PNVIBA diblock are dehydrated, resulting in more densely packed aggregates [51,52,53]. Although PNIPAM_128_-*b*-PNIVBA_93_ has a longer chain length than PNIPAM_128_ homopolymer, it shows smaller *D*_h_ than PNIPAM at higher temperatures than LCSTs, showing the unique thermo-responsive behaviors of PNIPAM-*b*-PNVIBA diblock copolymer. The zeta potential measurement of the PNIPAM-*b*-PNVIBA diblock copolymer shows a good agreement with the nanoparticle formation tendency observed through DLS, indicating that they are well dispersed in water media (Appendix A).

Variable temperature ^1^H NMR studies were performed in D_2_O solvent over a range from 25 to 55 °C for both PNIPAM_128_-*b*-PNIVBA_93_ diblock copolymer and PNIPAM_128_ homopolymer solution. (Figure 5) The signals of characteristic peaks, specifically the backbone ^1^H peak of PNIPAM at 2.0 ppm (indicated by the blue arrow) and isopropyl ^1^H peak PNVIBA at 2.1 ppm (indicated by the red arrow), were selected to observe the temperature drive transition. Initially, we expected that the PNIPAM peak would shrink faster than the PNVIBA peak because the LCST of PNIPAM is lower than that of the PNVIBA. However, the two peaks gradually collapse, and finally, both peaks are completely undetectable above 40 °C, which is a higher temperature than the LCST of PNVIBA. This result suggests that the thermal transition of PNIPAM and PNIVBA segment occurs in a cooperative manner rather than independently. This thermo-responsive character is expected from the result of strong interactions between PNIPAM and PNVIBA chains.

To confirm the self-assembled structure of the PNIPAM-*b*-PNVIBA diblock copolymer, scanning electron microscopy (SEM) analysis was carried out (Figure 6). The samples were prepared at 35 °C and 50 °C, which exceeds the LCST of PNIPAM and PNVIBA, respectively. At 35 °C, PNIPAM_128_-*b*-PNVIBA_93_ diblock copolymer is expected to form small spherical particles (<100 nm), but at 50 °C, the diblock copolymer particles collapse and form a larger cluster by aggregation. These SEM images are in good agreement with the DLS analysis, showing a mean diameter of 85.2 nm at 35 °C and 171.4 nm at 50 °C.

Based on the UV-visible, DLS, variable temperature NMR, and SEM analysis results, we propose the following step-wise sequential LCST behavior of PNIPAM-*b*-PNVIBA di-block copolymers as illustrated in Figure 2. It seems that the interaction between the PNIPAM and PNVIBA segments leads to a cooperative aggregation, and the polymer chains begin to collapse and form small aggregates, but further aggregation is stumbled by the PNVIBA segments. However, larger aggregates composed of clusters of small aggregates are formed as the temperature approaches the LCST of the PNVIBA chains.

## 4. Conclusions

Thermo-responsive diblock copolymer was synthesized via “Switchable” RAFT polymerization, a protocol for the controlled radical polymerization of monomers with different reactivities. Chain extension of the PNVIBA block from the PNIPAM macro-RAFT agent produced PNIPAM-*b*-PNVIBA diblock copolymers in a controlled manner. PNIPAM-*b*-PNVIBA diblock copolymer shows thermo-responsive behavior distinct from PNIPAM. A single LCST close to the LCST of PNVIBA was observed with the aqueous solution of the diblock copolymer, indicating that the interaction between the PNIPAM and PNVIBA segments of the block copolymer leads to cooperative aggregation during the self-assembly induced phase separation in the solution. Above the LCST of the PNIPAM block, the polymer chains began to collapse, forming small aggregates, but further aggregation stumbled due to the PNVIBA segments. However, larger aggregates composed of clusters of small aggregates formed when the solution temperature approached the LCST of the PNVIBA block. The thermo-responsive properties of PNIPAM-*b*-PNVIBA diblock copolymers may find various applications in drug delivery, tissue engineering, nanoreactors, and thermo-responsive sensors.

## Data Availability

Data are contained within the article and Appendix A.

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
