# Peer review of "Synthesis and Thermo-Responsive Behavior of Poly(N-isopropylacrylamide)-b-Poly(N-vinylisobutyramide) Diblock Copolymer"

_polymers, 2024, doi:10.3390/polym16060830_

Round 1
Reviewer 1 Report (Previous Reviewer 1)
Comments and Suggestions for Authors
The paper presents intriguing findings on a novel thermo-responsive diblock copolymer comprising poly(N-isopropylacrylamide) and poly(N-vinylisobutyramide). While the paper is well-organized, it lacks an adequate explanation of the copolymer's thermoresponsive behavior.
I find it challenging to comprehend why the authors assert, "Above the LCST of the 323 PNIPAM block, the polymer chains began to collapse, forming small aggregates, but further aggregation was stumbled due to the PNVIBA segments. However, larger aggregates composed of clusters of small aggregates formed when the solution temperature approached the LCST of the PNVIBA block." The dynamic light scattering (DLS) data does not seem to align with the behavior outlined in Scheme 2. It would be beneficial to incorporate temperature ranges in Scheme 2 to provide clarity.
In conclusion, it is crucial to articulate the significance and purpose of this research.
Comments on the Quality of English LanguageMinor editing of English language required.
Author Response
Thank you for your comment. We added the temperature range in Scheme 2. DLS data in Figure 4 (Hydrodynamic Diameter vs Temperature) clearly show the size change (Hydrodynamic Diameter) of the block copolymer coils and their aggregates as the temperature increased. The SEM images in Figure 6 also show the size change of the block copolymer coils and their aggregates. In addition to these data, the variable temperature NMR analyses support our assertion for the LCST behavior of the block copolymers.

Reviewer 2 Report (Previous Reviewer 2)
Comments and Suggestions for Authors
The authors revised their work according to the suggestions.
Author Response
Thank you for reviewing the manuscript.
Reviewer 3 Report (New Reviewer)
Comments and Suggestions for Authors
In reference #11 one is asked if the age of poly(NIPAM) is over? The authors answer that it is not yet, there is still a lot to explore. Indeed, the system studied by the authors is quite interesting: blocks of structural isomers with different LCST in one copolymer – it was very interesting to follow the changes in thermoresponsive properties. I believe that the work is free of serious flaws and will be of interest to readers of Polymers. Minor comments include:
1. For synthesized polymers, it is necessary to provide the theoretical molecular weights, for example, in Table 1.
2. For experiments to determine the LCST, please indicate the polymer concentration and wavelength of light. For DLS experiments, indicate the heating rate and whether there was thermostating at each temperature, and if so, for how long. Indicate pH during the measurement of zeta potential.
3. The TGA and DSC data presented in the Supplementary materials are not discussed.
4. The last sentence in the Conclusions could be more cautious (indicating the potential for use) or supported by specific [references] to practical applications.
5. In reference #12, there is an extra number, and references 30 and 37 are duplicates.
6. A question for the authors (does not require experiments): With regard to Figure 4, did the authors observe the change in Dh during cooling?
Author Response
Thank you for your comments. We revised the manuscript as suggested. The detailed responses are described in the attached response letter.

This manuscript is a resubmission of an earlier submission. The following is a list of the peer review reports and author responses from that submission.
Round 1
Reviewer 1 Report
Comments and Suggestions for Authors
The paper titled "Synthesis and Thermo-responsive Behavior of Poly(N-isopropylacrylamide)-b-Poly(N-vinylisobutyramide) Diblock Copolymer" describes the synthesis of a diblock copolymer using reversible addition-fragmentation chain transfer (RAFT) polymerization. The thermo-responsive properties of this copolymer were studied, particularly focusing on LCST. The paper discusses the characteristics of this copolymer in comparison to poly(N-isopropylacrylamide) (PNIPAM) and poly(N-vinylisobutyramide) (PNVIBA) as well as the effects of their interactions. Nevertheless, the paper presents interesting results, the paper can be accepted only after major revision. The novelty of the results should be exactly explained. The following issues should be clarified.
The introduction needs substantial improvement. It lacks clarity regarding the choice of N-vinylisobutyramide for copolymer synthesis and the expected properties. More information on the application of such systems with LCST should be provided. I recommend referencing prominent reviews on the topic, such as https://doi.org/10.1002/tcr.202300217 and https://doi.org/10.1007/12_2010_57
The paper should propose and illustrate hypothetical molecular mechanisms for the behavior of copolymer fragments at different temperatures.
Zeta potential measurements of the samples at various temperatures should be included to enhance the discussion.
The expected behavior of the copolymer in various pH and buffer solutions should be discussed in detail.
The addition of physicochemical properties of the newly synthesized sample, such as Thermogravimetric Analysis (TGA) and Differential Scanning Calorimetry (DSC), would be beneficial.
An appropriate discussion should be included regarding the advantages and disadvantages of the prepared sample when compared to other polymers with LCST.
Comments on the Quality of English LanguageMinor editing of English language required.
Reviewer 2 Report
Comments and Suggestions for Authors
This manuscript reports the preparation and characterization of Poly(N-isopropylacrylamide)-b-Poly(N-vinylisobutyramide) diblock copolymer by RAFT polymerization. The results show that the LCST of block copolymer is located at a temperature (36.9 °C) between the LCST of its homopolymers including PNIPAM (35.2 °C) and PNIVBA (39 °C, based on previous reports). It is quite easy to expect this combinational effect to be achieved by diblock copolymerization of NIPAM and NIVBA. However, the manuscript is a routing work. It lacks novelty and does not meet the standard of the Journal of “Polymers”. Therefore, the acceptance is not recommended. In addition, there are several issues to improve the quality of the work as following,
1. There are some typos and grammatical mistakes in the manuscript and need to be refined thoroughly. For example; on Page 1 Line 37, what means “LCST polymers”, Page 1 Line 40, what means “thermo-responsive transition”, at scheme one the degree sign (°) should be changed to superscript.
2. In the Introduction, the significance of the present research is absent; please declare the consequence of current results over previously reported. Besides, the references are approximately old, and recent works in this field should be considered in the introduction.
3. In the experimental section, the polymerization conversion should be defined, particularly for PNIPAM-b-PNVIBA diblock copolymer, because it seems that just 30 percent of NVIBA (1.69 g) has participated in block copolymerization with PNIPAM macro-RAFT agent (0.51 g) based on the results of molecular weight. It is expected to achieve a block copolymer like PNIPAM128-b-PNVIBA270, if all NVIBA is copolymerized with the macro-RAFT agent.
4. It is suggested to prepare PNVIBA copolymer and other ratios of PNIPAM-b-PNVIBA diblock copolymers together with their characterization to improve the significance of the work.
5. The utilized method and equation for measuring the molecular weight of the PNIPAM-b-PNVIBA diblock copolymer based on NMR analysis is confusing. In addition, the obtained molecular weight from NMR analysis will be compared with those obtained from GPC.
6. The SEM images with different magnifications which contain more particles are necessary to confirm the results of DLS and temperature-induced phase transition of samples.
7. The Conclusion section should be a summary of the principal achievements with quantification results.
Comments on the Quality of English LanguageThere are some typos and grammatical mistakes in the manuscript and need to be refined thoroughly. For example; on Page 1 Line 37, what means “LCST polymers”, Page 1 Line 40, what means “thermo-responsive transition”, at scheme one the degree sign (°) should be changed to superscript.
Reviewer 3 Report
Comments and Suggestions for Authors
Yoon et al. present an interesting study on PNIPAM-b-PNVIBA diblock copolymers. Even though this study is based on NVIBA, which is a relatively unexplored thermosensitive unit, and thus this idea fills this gap in the literature, the manuscript is not complete. Experimental details on all the aqueous characterisation techniques are missing, thus making it impossible for someone to reproduce the work. Additional results should be included in the supporting information and the main manuscript to support the current data and claims. Also, better characterisation on the chemical structure should be performed, to support the existence of two different repeated units, namely PNIPAM and PNVIBA. Considering these, I do not recommend publication of this research paper in Polymers in the current state of the manuscript. Please see the attached document for comments on how to improve the manuscript.
